# An Overview of Hydrothermally Synthesized Titanate Nanotubes: The Factors Affecting Preparation and Their Promising Pharmaceutical Applications

**DOI:** 10.3390/pharmaceutics16050635

**Published:** 2024-05-09

**Authors:** Ranim Saker, Hadi Shammout, Géza Regdon, Tamás Sovány

**Affiliations:** Institute of Pharmaceutical Technology and Regulatory Affairs, University of Szeged, Eötvös u 6, H-6720 Szeged, Hungary; rnmsaker@gmail.com (R.S.); hadishammout2@gmail.com (H.S.);

**Keywords:** TNTs, hydrothermal treatment, temperature, alkaline medium, post treatment, pharmaceutical applications

## Abstract

Recently, titanate nanotubes (TNTs) have been receiving more attention and becoming an attractive candidate for use in several disciplines. With their promising results and outstanding performance, they bring added value to any field using them, such as green chemistry, engineering, and medicine. Their good biocompatibility, high resistance, and special physicochemical properties also provide a wide spectrum of advantages that could be of crucial importance for investment in different platforms, especially medical and pharmaceutical ones. Hydrothermal treatment is one of the most popular methods for TNT preparation because it is a simple, cost-effective, and environmentally friendly water-based procedure. It is also considered as a strong candidate for large-scale production intended for biomedical application because of its high yield and the special properties of the resulting nanotubes, especially their small diameters, which are more appropriate for drug delivery and long circulation. TNTs’ properties highly differ according to the preparation conditions, which would later affect their subsequent application field. The aim of this review is to discuss the factors that could possibly affect their synthesis and determine the transformations that could happen according to the variation of factors. To fulfil this aim, relevant scientific databases (Web of Science, Scopus, PubMed, etc.) were searched using the keywords titanate nanotubes, hydrothermal treatment, synthesis, temperature, time, alkaline medium, post treatment, acid washing, calcination, pharmaceutical applications, drug delivery, etc. The articles discussing TNTs preparation by hydrothermal synthesis were selected, and papers discussing other preparation methods were excluded; then, the results were evaluated based on a careful reading of the selected articles. This investigation and comprehensive review of different parameters could be the answer to several problems concerning establishing a producible method of TNTs production, and it might also help to optimize their characteristics and then extend their application limits to further domains that are not yet totally revealed, especially the pharmaceutical industry and drug delivery.

## 1. Introduction

Nowadays, nanoparticles have attracted huge attention in different domains of science and human life such as chemistry, energy, and medicine due to their special design and small size. Titanate nanotubes (TNTs) are one type of these novel materials, which was first prepared in 1996 by Hoyer [1] after organic carbon nanotubes were presented in 1991 with promising results in different areas. TNTs are described as a spiral-shaped type of inorganic nanoparticles which are originally derived from titanium dioxide (TiO_2_) as a raw material [2]. Their diameter typically ranges between 10 and 100 nm according to the applied preparation method, but it might reach even 400 nm in some cases [2,3]. This nano-size combined with tubular structure are unique properties which differentiate them from other types of nanoparticles (NPs), giving them higher surface area and cellular uptake [3,4]. Moreover, TNTs offer a distinguished profile of specifications that could be interesting in numerous fields, such as their hydrophilic nature, good wettability and biocompatibility [2,3,5], chemical stability [5], photocatalytic activity [5], excellent mechanical properties, and corrosion resistance [5,6], in addition to low toxicity [7,8,9,10]. Due to these remarkable specifications, TNTs have been successfully utilized in numerous fields with great outcomes and impressive performance (Figure 1) making them attractive candidates for investigation of their potentials in further disciplines [5,11,12,13]. Although some attempts have been made to investigate the possible application of these newly synthesized materials in biomedical research, few of them have focused on their application in the pharmaceutical field as drug carriers [5,14,15,16,17].

Since their first appearance, many methods have been proposed for the proper preparation of TNTs, such as sol gel template, template-assisted synthesis, electrochemical treatment, and hydrothermal treatment [2,5]. Based on the literature data, it was clearly concluded that the final properties of the manufactured materials, especially their morphology, diameter, and crystal structure, are highly dependent on the method of preparation, which will subsequently affect their application field. For example, the diameter of nanotubes differs according to their preparation method, as electrochemical treatment would result in self-organized but large nanotubes (the typical diameter is >25 nm, often ≈100 nm). This large diameter is not convenient for biomedical uses because nanoparticles with this size will be easily cleared from the circulation by the reticuloendothelial system (RES). On the other hand, the use of the templating method is also limited in obtaining small-size nanotubes, due to the limitation of the pore size of the mold [3,5]. In contrast, hydrothermal treatment would result in a smaller diameter (10 nm) along with possibly hundreds of nanometers in length, making them possible candidates for medical and pharmaceutical use [2].

The small diameter and long circulation half-life of hydrothermally synthesized TNTs are not the only advantages that this method could offer, so it is becoming one of the most popular methods for the preparation of TNTs. It is a simple process that involves the crystallization of a starting material from a highly concentrated alkaline solution under a specific set of parameters (temperature, time, pressure). Moreover, as water is used as a reaction medium and no organic solvents are involved in any step of the process, it can be considered as an environmentally friendly and safe procedure to apply in laboratories and in larger scale production. Hydrothermal synthesis also provides high yields and mass efficiency, which make it an attractive candidate for scaling up to industrial level [5,11,18]. However, although it is one of the most commonly used and discussed methods in the literature, it still generates controversies in the scientific community, as the published results are not totally in agreement. The differences in applied conditions/parameters during preparation makes the comparison of the results difficult, which could lead to conflicting results. Many theories have been proposed to explain TNTs formation. The most accepted one among authors using this method for the production of nanotubes involves several steps (Figure 2): breaking of Ti-O-Ti bonds after the dissolution of TiO_2_ precursor in a strong alkaline medium, formation of Ti-O-Na bonds resulting in lamellar sheets, and rolling of the lamellar sheets into nanotubes [19]. According to Tsai et al., the starting TiO_2_ precursor is converted into sheets through alkali treatment. These sheets mainly exist as an anatase phase that is more suitable for rolling to form nanotubes [20]. However, there is still a debate whether the formation of nanotubes occurs during the dynamic hydrothermal process [21,22,23] or only after exchanging sodium ions through acid treatment, which would lead to the scrolling of the precursor sheets [24,25,26,27]. The answer could highly depend on the reaction itself, as its conditions would significantly affect the resulting material, especially the properties/ratios of the starting precursors, sodium hydroxide (NaOH) concentration, reaction temperature, reaction time, mixing, and post treatment. Variations of these factors could affect the properties of the resulting nanotubes or even favor the formation of nanotubes with other nanomaterials such as nanosheets or nanoribbons. This could explain why it is always expected to have nanosheets that are not completely rolled up as byproducts of this preparation method [2,3].

Based on the previous discussion, the specifications of the prepared nanotubes are closely related to the preparation method and strongly sensitive to the variation of its parameters. Hence, there is an urgent need to determine the appropriate range of preparation conditions under which nanotubes are produced in addition to precisely understanding the changes that could occur outside these ranges, such as changing morphology or crystal structure.

Therefore, this review highlights the importance of conducting an extensive screening and precise analysis of the different parameters involved in hydrothermal synthesis and their impact on the produced material, which could be the key to justifying the conflicted observations reported in the literature and to gaining a better understanding as to how to optimize this procedure to obtain the targeted nanotubes.

This investigation is of crucial importance in establishing a reproducible, robust, and scalable method for producing TNTs with controlled features as an initial step to later upgrading their use to the pharmaceutical domain.

The main parameters discussed in this review can be classified as thermal (reaction temperature, post-treatment calcination), chemical (concentrated alkaline medium, acid washing), or even mechanical (stirring, agitation). The variation of these parameters could result in significant transformations, and controlling them will open the door for optimizing the production process and extending the applications of TNTs to new and innovative platforms. 

This review also discusses the most important attempts that have been made in the last two decades regarding the utilization of TNTs for therapeutic purposes as drug carriers.

## 2. Hydrothermal Synthesis Conditions

### 2.1. Reaction Temperature 

According to the literature, the temperature involved in the different stages of TNTs’ preparation and post-preparation treatment highly affect their properties such as morphology, specific surface area (S.S.A.), and crystal structure, which would therefore affect their behavior and subsequent application, especially those related to their photocatalytic activity [28]. 

The utilization of a sufficient temperature (<130 °C) would lead to the formation of Ti-O-Na and Ti-OH bonds after breaking Ti-O-Ti bonds in the starting nanoparticles. This would subsequently lead to the formation of lamellar TiO_2_ sheets because of the electrostatic repulsion of the charge on sodium. These lamellar sheets are considered an intermediate stage as they can scroll up to form the tubular structure. Treatment at higher temperatures results in granular particles rather than lamellar sheets. On the other hand, treatment at lower temperatures (<110 °C) results in thick plates which are not able to scroll and form nanotubes; thus, the extent of precursor conversion, the yield of the produced nanotubes, and the overall success of the procedure is determined by the treatment temperature [20].

Vu et al. suggested the same mechanism, as, at lower temperatures, the sphere-like particles and sheet-like structures will be dominant, but increasing the temperature would provide sufficient thermal energy to curl up nanosheets into tubular structures. This was first generated above 120 °C with the existence of some nanosheets that had completely transformed into tubes beyond 130 °C [21]. This was in agreement with the study of Sreekantan et al., who reported that temperature affects the morphology of the treated sample, as, at lower points (90 °C), the conversion of spherical particles to nanosheets will start, but the whole transformation would happen at higher temperatures (>110 °C) because, at this point, the additional thermal energy will lead to a sufficiently high surface energy to curl up the sheets [29]. In the same context, Ribbens et al. also reported that a low temperature (<110 °C) is not capable of forming nanotubes [30]. In addition, Viet et al. also demonstrated that a temperature lower than 115 °C is not enough for the formation of nanotubes. According to the study, the optimal range for successful preparation is between 115 and 180 °C, as a higher temperature could destroy the lamellar structure of TiO_2_; therefore, nanotubes will not appear [27]. Yuan et al. also reported that the optimal range of hydrothermal temperature is 100–150 °C [31]. According to their findings and others, increasing the temperature within this range would increase the yield of nanotubes [31,32], their crystallinity, and outer diameter [32].

Increasing the reaction temperature would also increase the S.S.A. up to 150 °C, but, interestingly, a higher temperature would lead to opposite results (decreased S.S.A.) [33]. Lee et al. agreed, indicating that at a temperature greater than 160 °C, enormous decrease in surface area and pore volume will happen as a result of the formation of thick titanate nanorods [34]. Similar to the previous results, high temperature treatment led to the formation of different structures other than nanotubes in several studies, such as nanoribbons (>160 °C) [31,35], nanofibers or nanobelts (>170 °C) [36], and nanorods (180 °C) [32]. 

Bavykin et al. also studied the effect of temperature on diameter/morphology, concluding that increasing it in the range of 120–150 °C could increase the average diameter of the prepared nanotubes, but a subsequent increase could lead to morphological change presented as non-hollow nanofibers [37]. Although increasing temperature would generally increase the number and length of the formed nanotubes, it would also result in a decreased internal diameter. It is well known that S.S.A. normally increases with increased temperature, but it is also in proportion to the internal diameter; therefore, the decreasing of internal diameter induces a decrease in the surface area of the produced nanotube [38].

### 2.2. Reaction Medium

As previously mentioned, the hydrothermal treatment method uses an alkaline medium which is mostly an aqueous solution of NaOH, and the concentration of this solution would affect the properties of the produced nanotubes, especially their morphology and surface area [39]. The full growth of nanotubes requires a highly concentrated solution of 10 M, as, below this limit, a large number of aggregates of unreacted powder, particles, and sheets will be observed. S.S.A. would also be affected and increases with increasing alkali concentration [21]. Too low (<5 mol/L) or too high (>15 mol/L) a concentration of NaOH will result limited formation of nanotubes. The optimal range for a high yield was found to be between 10 and 15 mol/L [31]. It was also found that NaOH concentration is the most important factor affecting morphology, and its effect is even stronger than that of temperature and reaction time. However, optimization of these three factors could be an effective tool to control/adjust the final shape [39]. In contrast, another study reported that, if a lower concentration of NaOH is utilized, a higher temperature could be used to fill this gap and obtain the desired tubular structure, suggesting that temperature could be the most critical factor affecting nanotubes formation [35]. 

In the same context, Guo et al. reported that NaOH concentration affects the scrolling of multi-layered nanosheets to form nanotubes, as low concentrations result in a weak driving force for scrolling, but extremely concentrated NaOH will interrupt the scrolling process [40].

Although NaOH is considered to be the most often used alkaline medium in hydrothermal treatment, other types of alkaline medium, such as potassium hydroxide, could also be applied. However, it was reported that using NaOH was more effective than potassium hydroxide in the enhancement of nanotube formation [41].

### 2.3. Reaction Time

The treatment time is a very important factor that gives the required time for nanomaterials to turn from a spherical shape into an intermediate phase (nanosheets) and finally into nanotubes. The complete transformation would require approx. 15 h [29]. It also highly affects the length of the resulting nanotubes, as a longer time would result in increased length [21]. Ma et al. reported that the optimal time period is about 24–72 h to obtain a high yield of TNTs, as a shorter duration would increase the ratio of unreacted TiO_2_ particles [36], while Elsanousi et al. suggested a different opinion. Their results were in agreement with the study of Sreekantan et al. [29] that a short treatment (5–15 h) is enough to form pure nanotubes, as a longer duration would result in nanoribbon formation beside nanotubes, while pure nanoribbons are observed after very long treatments (48 and 72 h). However, the different applied temperature could be a reason for these conflicting results, as it is also deeply involved in this transformation. It is worth mentioning that nanotubes did not transform into nanoribbons at a low temperature, even if the duration of treatment was long, indicating that this transformation depends on several critical conditions in addition to the duration of the reaction, such as temperature and even pressure [42].

Guo et al. indicated that the thickness, width, and length of TNTs increase with the prolongation of process time [40]. Increasing the treatment duration leads to a growing amount of intermediate phase (nanosheets) through a dissolution–reprecipitation process and an increased tendency of curling and forming nanotubes. However, no further increase in their length was observed after 24 h of treatment [43]. In the same context, Ranjitha et al. indicated that the formation of lamellar fragments (the intermediate phase in TNTs preparation) would be the result of a short reaction time, and, as the reaction continues, the tubular structure will be obtained. They also highlighted the effect of reaction time on TNTs crystallinity and reported that the amount of rutile phase would increase with the increasing reaction time [44]. Elsanousi et al. also indicated that the crystallinity of the final product might be improved by increasing the duration of the treatment [42].

According to the previously mentioned results, it is widely accepted that increasing the duration of heat treatment would improve crystallization, but at the same time it would negatively affect the surface area and therefore highly influence the related surface applications such as photocatalytic activity. From this point of view, if photocatalytic activity is the main target of the TNTs production, a compromise between surface area and crystallinity should be obtained to achieve their desired synergistic effect [45]. 

The optical properties would also be influenced by the duration of treatment. It was reported that the absorption edge of TNTs has a red shift (to a higher wavelength) with longer reaction time due to the modification of the TiO_2_ band gap, concluding that increasing hydrothermal reaction time will decrease the band gap energy and increase the wavelength of the absorbed light [44].

### 2.4. Reaction Pressure

The effect of pressure on the formation of nanotubes was not intensively investigated. It was reported that the pressure in addition to the volume of solution inside the used autoclave will affect the crystallinity of the resulting nanotubes. This pressure is highly related to the filling fraction and to the thermal expansion of different volumes of the liquid. It was also reported that more pressure is needed in the case of nano powder to form crystalline nanotubes in comparison to micro powder [46]. Tsai et al. indicated that the pressure of saturated NaOH solution in the sealed autoclave would only impact the formation of TNTs at temperatures higher than 150 °C, as the differences between pressure values at temperatures lower than 150 °C would be minor [47]. Morgan et al. studied the effect of pressure on nanotube formation at a higher temperature range (up to 220 °C) and concluded that high pressure (accompanied with lower NaOH concentration) was not promoting nanotube formation as expected, so they excluded the pressure effect as an influencing factor [35]. 

### 2.5. Stirring, Agitation and Sonication

Mechanical forces should also be considered while listing the factors affecting the preparation of TNTs. For example, a stirring process could shorten the time needed for the transformation of TiO_2_ into nanotubes as it would increase the contact between the precursor and NaOH, while, in a static condition without stirring, only a small amount of TiO_2_ nanotubes would form. Thus, stirring clearly affects the production yield [21]. Agitation could also affect the morphology/surface area of the prepared TNTs, as it could accelerate the dissolution process of the precursor during preparation, thus affecting the subsequent nucleation after supersaturation [48].

Ma et al. indicated that the process of sonication and its power affect the morphology of nanomaterials during hydrothermal treatment. Sonication with enough power promotes the breaking of Ti-O-Ti bonds and sodium ions intercalating into the titania lattice; therefore, increasing the sonication time will result the transformation of the spherical particles into a rod configuration. These nanorods will continue to grow during hydrothermal treatment, extend their length, then form the tubular structure after washing with acid [49].

### 2.6. Starting Materials

It was previously reported that impurities and different sources/types of starting materials have no effect on nanotube formation [21]. Any type of TiO_2_ could be used as a precursor for the preparation of TNTs. However, the properties of the used precursor would have a high impact on the resulting materials. For example, the particle size of the TiO_2_ starting material could affect the morphology/surface area of the produced nanomaterials, as smaller TiO_2_ starting particles will have a higher solubility in the preparation solution and this will result in a higher concentration of Ti ions, leading to a homogenous nucleation after the supersaturation is achieved. If a lower concentration of Ti ions is present due to the larger particle size of the starting material, heterogenous nucleation is dominant [48]. Yuan et al. concluded that surface area would be influenced by the particle size of the raw materials, as finer particles will lead to a higher surface area, while yield is not significantly affected [31].

The smaller particle size of the starting material could also result in a faster formation of TNTs [41]. Ma et al. also reported the importance of the precursor’s particle size, as smaller ones could effectively form TNTs during a combined sonication–hydrothermal process. However, precursors with larger particles would form sheets with rolled edges which cannot turn into the tubular shape, probably due to the steric hindrance of the larger particles [49].

In addition, the differences in TiO_2_ precursor structure will also affect the formation of TNTs and their resulting properties. The type of the starting material would not highly affect the diameter and the thickness of the walls, but thermal stability would differ according to it. The anatase precursor presents TNTs with the highest thermal stability and preserves their tubular structure up to 400 °C, while an amorphous precursor presents TNTs with less thermal stability. This could be attributed to the defective sheets/foils formed from these precursors. Crystalline precursors originate fewer defective sheets compared to the ones originated from amorphous precursors. These defects make the occurrence of the rolling up process difficult, so more nanosheets were detected compared to the case of crystalline precursors [50]. Surprisingly, Yuan et al. reported that amorphous TiO_2_ as a precursor did not lead to TNTs formation [31].

The crystalline phase of the used precursor will also affect the speed of TNTs formation, as anatase (a less stable phase with higher energy) reacts with NaOH faster than rutile during preparation [49]. Seo et al. also found that the anatase phase has a greater tendency for continuous elongation in a specific direction during preparation, which makes it more effective in TNTs formation than the rutile phase [38]. Moreover, the concentration of the starting material (TiO_2_ precursor) is also a crucial factor to take into consideration, as increasing it apparently increases the length of the prepared nanotubes while the diameter is still the same. This systematic change in the aspect ratio reflects the increased density of the initial nucleation sites [51]. 

It could be concluded from the above-mentioned results that the morphology, surface area, phase transformation, thermal stability, and yield of TNTs highly differ according to the properties of starting materials, in addition to other parameters such as reaction time/temperature [41]. It should also be mentioned that titanium metal could also be used as a precursor for the preparation of TNTs in addition to crystalline and amorphous TiO_2_ [40].

### 2.7. Molar Ratio

The molar ratio of starting materials (TiO_2_ and alkali) will also affect the diameter and surface area of the resulting nanotubes. Increasing the TiO_2_:NaOH molar ratio leads to a decrease in surface area and an increase in the average diameter of the nanotubes. A slow convection in the autoclave during preparation could be responsible for these differences in morphology [37]. The molar ratio will also play a significant role in gaining the desired morphology, as a higher NaOH:TiO_2_ molar ratio means the existence of excess sodium ions that provide high surface energy, leading to the scrolling-up of the structure into nanotubes and a smaller outer diameter, while a lower molar ratio leads to the deposition of nanoparticles on the wall of the rods rather than the development into a tubular structure [29].

According to the literature, the chemical properties of prepared nanotubes would also be affected by the amount of NaOH and the Ti/Na ratio. Studies have shown that increasing the amount of NaOH will lead to a decreased number of Lewis acid sites on the surface. Furthermore, surface acidity of TNTs (Brönsted acid sites) also depends on the Ti/Na ratio, since increasing NaOH concentration to a sufficient quantity will enable the formation of Ti-O-Na, which transforms into Ti-O-H after washing with water. Basically, the surface of both anatase and rutile TiO_2_ has only Lewis acid sites, but increasing the Ti/Na ratio in addition to scrolling will generate more Brönsted acid sites due to exfoliation of the Ti-O planes. This surface acidity is nowadays a critical factor in several applications such as heterogeneous catalysis. To preserve this acidity in TiO_2_ nanostructures, the best Ti/Na ratio is 7 [52]. The Ti/Na ratio also affects morphology, as different structures could be obtained by modifying it, such as nanofibers, nanotubes, and nanowires [52].

## 3. Post Treatment

### 3.1. Acid Treatment

As was previously mentioned, the various results have suggested that formation of TNTs and obtaining the tubular structure do not only depend on the alkaline synthesis but could be highly dependent on the washing step after it. Some reported results indicated that hydrothermal treatment is not enough to form the tubular structure, but an additional washing step with acid should be involved in the preparation procedure to transform the disordered intermediate phase into the desired geometry, due to a gradual Na^+^/H^+^ substitution [47]. This critical point of disagreement was further discussed by Nguyen et al., who concluded that hydrothermal treatment may not be sufficient alone to form the tubular structure for two reasons: first, the existence of a high sodium content if the washing process with water was not completed; second, washing with acid could serve as an assisted condition if the reaction condition was not sufficient for the formation of nanotubes [53].

In addition to the proposed theory that acid washing is necessary to obtain the desired tubular morphology, pH variation of the applied washing solution may cause crystal-structure transformation through simple rearrangement and could also lead to surface area variations. Nevertheless, it could also negatively affect morphology, as increasing the acidity of the washing solution would result in defective nanotubes caused by formation of anatase TiO_2_ phase on some spots of the tubes. Also, extreme pH values could lead to coagulated particles or plates [47].

The previous results were in agreement with the study of Nguyen et al. who reported that the pH value of the washing solution will affect the morphology and thus the surface area of the produced TNTs. Decreasing pH from 9 to 2 resulted in more successful tube formation and, subsequently, a higher area, but when pH further decreased from 2 to 1 a decreased surface area could be noticed because of the low yield of nanotubes and appearance of nanosheets under this highly acidic condition. It is also worth mentioning that TNTs washed at pH = 0.5 were completely dissolved in the washing solution without any residual amount to be examined. In conclusion, TNTs washed at pH 1.6–4 showed the best appearance of tubes and the highest amount of tubular structures, while TNTs washed at pH = 2 showed the highest surface area, pore volume, and pore size [53]. 

According to other studies, acid treatment could also affect the size and crystallinity of the resulting nanotubes and result in a smaller average size and the appearance of crystalline rutile peaks. The untreated samples were only amorphous by examination [27].

It is of crucial importance to mention that the washing step mainly affects the sodium content of TNTs; therefore, increasing the number of washings could eliminate the sodium ions from the surface, although their presence is very important to preserve the regular structure. This could explain why washing with a high excess of liquid produces nanotubes with irregular shapes and thinner walls [24]. It also should be mentioned that the applied concentration of acid should be carefully determined in the post treatment, as some researchers have pointed out that highly concentrated acid (>0.01 M) could destroy the tubular structure, as it could break the Ti-O-Na bonds and lead to surface irregularity, confirming the previous result that the existence of sodium ions on the surface has a significant role in preserving the tubular structure of TNTs [54]. Nada et al. reported the same results, which confirm that only low-acid concentrations should be used during the washing step, otherwise a damaged structure could result [25].

However, Poudel et al. used a wider range of concentrations, suggesting that the optimal concentration of washing acid is between 0.5 and 1.5 M to obtain pure nanotubes with a high yield. According to the study, concentrations below 0.5 M were not capable of removing sodium impurities, while concentrations above 2 M could damage nanotubular structure, leading to formation of clumps. The type of the applied acid was not significant, as treatment with nitric acid gave similar results [46]. This agreed with the findings of Turki et al., who also indicated that the use of a highly concentrated acid (above 2 M) would result in formation of clumps of irregular particles and damaged morphology. The complete removal of sodium ions by acid treatment did not significantly modify the morphology. However, some modifications can be noticed, such as fewer crystalline walls, a rough surface, larger pores, and a higher surface area [19]. 

Porosity also appeared to be affected by the concentration of acid. It would dramatically increase with higher concentrations up to 0.2 M. However, decreased porosity was detected with further increases, as fast elimination of electrostatic charges above 0.2 M would lead to sheets folding and transforming into granules. A decreased size of nanotubes could also be noticed. In conclusion, the products from a concentrated-HCl washing solution consist of granules rather than tubes [20].

Reducing the sodium content of TNTs by acid washing would also lead to a different response to temperature (different phase evolution), as TNTs with a lower Na content (the most effectively washed TNTs) would turn into the crystalline phase at a lower temperature. Furthermore, they would also lose their tubular structure due to sintering earlier compared to TNTs with high sodium content, confirming the theory that high sodium content provides higher thermal stability [23,55]. These findings are well correlated with those reported in the literature that the thermal stability of TNTs improves when washed under less acidic conditions [53]. Ribbens et al. reported that Na-TNTs exhibit higher thermal stability due to the stabilization effect of intercalated sodium ions, so no phase transformation can occur as a response to the extreme temperatures involved in many procedures such as calcination [30]. For example, Na-TNTs were found to preserve their tubular structure after calcination at 400 °C, but a higher temperature will lead to a partial or complete loss of structure (500 and 600 °C, respectively). In contrast to Na-TNT, H-TNTs completely lose their tubular structure at a calcination temperature starting from 500 °C. This result confirms the previous ones suggesting that better stability of nanotubes’ morphology could be obtained with the presence of sodium ions, and this is only true at low temperatures; whereas, at higher temperatures, this effect was not detected [19].

There are also other differences reported in the surface properties of Na-TNTs and H-TNTs, as this is smooth in the case of Na-TNTs but obviously not in case of H-TNTs. The ion exchange during acidic post treatment leads to a smaller inter-layer distance as a result of replacing large sodium ions with small protons. It also leads to differences in the dimensions of nanotubes and the surface area [30]. Low sodium content was found to present a higher S.S.A. [23]. In the same context, Nakahira et al. reported that increasing the concentration of washing acid will increase the S.S.A. of the washed nanotubes [55]; thus, a higher specific area of the H-nanotubes means more active sites to react/bind, thus making it a more effective use in several applications [30]. 

It was also reported that sodium content also affects many functions of TNTs, as untreated samples (washed with water instead of acid) showed higher photocatalytic activity [41].

### 3.2. Thermal Treatment (Calcination)

According to the information reported in the literature, the temperature range (200–600 °C) is the range where the most important structural transformations probably happen. Annealing at high temperature (300 °C) is important to improve the crystallinity of TNTs and, thus, their performance in future applications such as photocatalysis. However, preserving the tubular structure during this treatment is highly important, as an extreme annealing temperature could lead to sintering (at 500 °C) or complete collapse of nanotubes and nanoparticles formation (at 600 °C) due to dehydration of inter-layered OH groups and reduction of the number of hydrogen bonds. It could also cause an irregular shape (>600 °C) and a change of the anatase phase to rutile [29,56]. 

Similar findings were reported by Kiatkittipong et al. that calcination at moderate temperature (200 °C) is of crucial importance as it will lead to the appearance of an anatase phase and elimination of moisture that existed between titanate layers, but a high calcination temperature (500 °C) would lead to complete collapse of TNTs with elongated and irregular particles, which is confirmed by the decrease in S.S.A. and loss of pore volume [57]. This was in agreement with the study of Lee et al., who confirmed that an extremely high temperature used in calcination would negatively affect surface area and pore volume [34], and with the study of Yuan et al., who reported that nanotubes will be stable only at a calcination temperature lower than 400 °C, while they would be sintered and turned into nanorods after calcination at 540 °C [31]. Ma et al. also indicated that the tubular structure can remain intact at a calcination temperature below 450 °C, as a higher temperature would completely destroy the tubular structure [49].

Besides morphology/surface area changes, calcination would also affect the crystalline phase. It was reported that at a calcination temperature range of 200–800 °C the structure was primarily anatase, while rutile started to appear at 900 °C [58]. Another study suggested a similar discussion but at a relatively different temperature. Nanotubes mainly existed as an anatase phase below 650 °C, preserving their morphology. However, a rutile phase began to appear at a lower temperature (550 °C) compared to the previous study (900 °C). Further thermal treatment from 650 °C to 800 °C will accelerate the transformation to the rutile phase, and, therefore, the product mainly showed a rutile phase over 800 °C. However, at this high temperature, the tubes were destroyed and aggregated together to form rod-like particles [26]. In contrast, another study suggested that a temperature above 800 °C would turn nanotubes into aggregated nanoparticles but the anatase phase would be dominant [43]. 

In conclusion, this extreme thermal treatment would modify the characteristics of nanotubes such as morphology, surface area, and crystallinity and, consequently, the related application possibilities [29]. For this reason, the targeted application should be decided earlier to the production process so that the synthesis conditions can be adjusted to obtain the appropriate/balanced specifications for the requested application. For example, although S.S.A. is of crucial importance for photocatalytic activity, as discussed earlier, this still depends on the crystallinity of the prepared nanotubes [58]. An improvement in photocatalytic activity was observed after increasing the calcination temperature up to 400 °C, despite an approximately 30% decrease in surface area, which could be explained by the increase in the percentage amount of anatase. However, an extremely high temperature (above 500 °C) would lead to the loss of tubular structure and to a decrease in photoactivity irrespective of the improved crystallinity [57]. The factors influencing the properties of the nanotubes are summarized in Figure 3.

## 4. Stability of the Prepared Nanotubes

As TNTs have promising potential to be used in multiple applications, their fragile structure should be taken into consideration, as it could undergo numerous changes during handling/processing. For example, TNTs could be easily broken up due to mechanical forces such as stirring or ultrasonication, resulting in shorter nanotubes [10,37]. Also, extremely high-temperature treatments used in many procedures could result in changing the surface chemistry and reducing the number of OH groups on it. Later, this could negatively affect their reactivity [59], photocatalytic activity, and other possible surface modifications, as a result of decreased Brönsted/Lewis acid sites and diminished surface OH-groups, which are the main sites of reaction [30,52]. In addition to that, pH could affect the stability of TNTs as they are not stable under acidic conditions. It was reported that treatment with 0.1 mol/L H_2_SO_4_ for 5 days at room temperature led to a deformation of TNTs, evidenced by thinner walls and corroded surfaces. Two months later, the tubular structure had completely disappeared, while, in pure water and basic solution, minimal morphological changes were observed, indicating the stability of TNTs in these media. This was explained by favoring the layered sodium titanate phase with high pH and high sodium ion levels, which stabilize the multi-layered wall TNTs [59]. On the other hand, the explanation of TNTs transformation in acidic media includes partial dissolution of nanotubes, releasing Ti, which is then crystallized to nanoparticles. This procedure depends on the acid type and decreases in the order H_2_SO_4_ > HNO_3_ ≈ HCl, as a higher release of Ti during dissolution resulted in faster transformation [59].

## 5. Applications in the Pharmaceutical Field

As previously mentioned, TNTs present a wide range of specifications that could have great importance on their usefulness in different applications. For example, their hydrophilic properties and good wettability, due to their partially hydroxylated surface, make them suitable candidates to use in the biomedical field, such as in dental implants, orthopedics, and cardiovascular stents [14,15,16,17]. However, agglomeration could appear as a result of the presence of hydroxyl groups on their surface [2], and this should be taken into consideration during the preparation process.

TNTs also have a higher S.S.A. compared to the traditional type of TiO_2_ [52]. This vast surface created by the hydrothermal alkali treatment provides higher activity compared to precursor nanoparticles and could be an advantage for various uses such as catalysis in chemical reactions [20]. It is also an available option to tailor the surface chemistry by functionalizing it with different molecules or replacing Na^+^/H^+^ ions on it with other ions such as Cu^2+^/Ni^2+^ and Mg^2+^. This would help to control/adjust TNTs properties and provide them with secondary characteristics and new behaviors other than the original ones, which could be the key to overcome some of their problems and extend their application to further disciplines [60,61,62,63]. 

These impressive characteristics are gradually being used in the biomedical field and even in the field of animal treatment. The highly specific area of TNTs was considered as an advantage in preparing drug delivery systems specified for animals for the first time. This was performed by physical absorption of enrofloxacin hydrochloride, and controlled drug release was successfully obtained [64]. In addition, Papa et al. reported that the tubular shape and nano-size of TNTs facilitate their cell uptake and present them as promising agents for DNA transfections in the cardiovascular field [9]. Table 1 contains more examples of the possible uses of TNTs in pharmaceutical field.

According to the previously mentioned specifications and uses, TNTs have proven to be an efficient tool in several fields. However, no serious attempts have been made to use them as novel carriers of therapeutic molecules. Their pharmacokinetics in the human body have not yet been totally revealed, which could be one of the major reasons for not fully exploring their potentials in pharmaceutical applications. Furthermore, their safety/toxicity studies still present conflicting results. Some studies have reported their safe use without showing any toxic effects [7,10,80]. Others even displayed positive outcomes such as increased cell viability [8], enhanced proliferation, and bone cell adhesion [81], probably due to their porous structure allowing the flow of nutrients and blood [82], while other studies reported their toxic effects and classified them as more toxic than other nanomaterials such as nanorods, nanoribbons, and nanowires [83,84]. However, the potential utilization of TNTs in health-related applications is supported by the promising results obtained by numerous studies which investigated their cytotoxicity using different cell lines (intact and cancerous) with different concentrations (range from 1 µg/mL to 5 mg/mL) and different times of exposure (from short treatment of 120 min to long exposure of 7 days) [7,8,10,85]. Even surface-modified TNTs with several molecules (silane, stearate, polyethyleneimine, acrylic acid-polyethylene glycol copolymer) have presented similar results which could promote the shifting of TNTs (bare and surface-modified) to the health domain [9,10,61].

Based on these results, researchers have started to give more attention to TNTs and to consider them as possible carriers for drugs/biological molecules. For example, Ray et al. prepared nano-conjugates using HTNTs with cytochrome C. These conjugates displayed enhanced peroxidase enzymatic activity and could be of great importance for cancer treatment in the future, as they could work as caspase activators and apoptosis initiators [86]. Moreover, TNTs have proven to be a promising tool to apply in oncology. Hydrothermally synthesized TNTs have been investigated for their possible use as radiosensitizers in radiotherapy. They appeared to increase the accumulation of cells at the G2/M checkpoint and decrease DNA repair efficiency, making cancer cells more sensitive to radiotherapy [87]. Similar findings were reported with TNTs prepared by anodization, as they were able to reduce the number of vital cancerous cells upon a low dose of UV irradiation [88]. These results were reported with bare TNTs without being loaded with any therapeutic agents; therefore, researchers expect to have a stronger effect if the combination with antitumor drugs is successfully accomplished. In this context, Wang et al. reported the possibility of using TNTs as an antitumor delivery system for doxorubicin to pancreatic cancer cells. This was achieved by loading the drug onto the surface of these nanotubes [89]. Furthermore, TNTs present a versatile method of carrying docetaxel on their surface after grafting, and they were able to deliver it locally to prostate tumors in addition to enhancing the response to irradiation by delaying tumor growth [90]. TNTs have also been investigated for their ability to graft optical imaging probes (OI) for diagnostic purposes and photodynamic therapy (PDT) [91].

TNTs were also used as carriers for other drugs such as ibuprofen and dexamethasone, and it was found that the release rate of these drugs could be regulated through modifying surface chemistry, with molecules having different properties such as a different polarity [92,93]. Modifying surface chemistry could also play a significant role improving drug loading with several drugs (curcumin, methotrexate, silibinin, ibuprofen), decreasing toxicity and achieving sustained release [94,95].

However, until now, few attempts have been made to incorporate those newly synthesized materials into API-composites and then into final dosage forms. For example, Sipos et al. investigated the impact of TNTs’ incorporation with different drugs, diltiazem hydrochloride, diclofenac sodium, atenolol, and hydrochlorothiazide, indicating that TNTs–drug composites appeared to have superior processability compared to pure active pharmaceutical ingredients (APIs), such as better flowability, compactibility, compressibility, and even post-compressional properties of the composite-containing tablets (greater hardness and tensile strength). In the same context, it is of crucial importance to mention that these advantages are closely related to the success of the incorporation process and the type of the created interaction between both entities, which strongly depends on the drug characteristics [6,96,97]. Accordingly, TNTs could be implemented to modify drug release from their composites based on the strength of the created drug–TNTs association [97,98].

It is also worth mentioning that successful incorporation could be a useful tool to shorten preformulation studies regarding the tableting process, as the unique properties of those novel carriers will be predominant, so the composites will behave similarly regardless of the used APIs [6].

## 6. Future Perspectives, Concerns, and Challenges

Hydrothermal synthesis may be the best choice to produce TNTs for pharmaceutical applications due to its numerous advantages over other techniques, mainly the long circulation half-life due to their small diameters being undetected by RES [2]. However, this intended shifting is not an easy mission to accomplish due to several problems TNTs are facing. For example, their hydrophilic nature is supposed to reduce their accumulation in tissues, but it is also expected to hinder their absorption through the gastrointestinal tract (GIT) [61]. Moreover, their tubular morphology is considered as an advantage boosting their cell internalization [2], but this could also increase their cytotoxicity [83]. Based on this, their properties should be carefully balanced to utilize their benefits and avoid their drawbacks. In this context, functionalization appeared as a golden choice to adapt TNTs characteristics and shape their features to suit the intended application. Ranjous et al. have modified the surface of TNTs with hydrophobic moieties (trichloro(octyl) silane and magnesium stearate) to enhance their GIT absorption after oral administration [61]. Saker et al. have also highlighted the importance of surface modification by fixing a carboxylic arm on the surface of TNTs to serve as a connecting bridge for several molecules. This approach could be useful for optimizing the surface properties to achieve multiple targets, such as reducing toxicity, enhancing water dispersibility, etc., or for loading therapeutic/diagnostic agents (APIs, biomarkers, probes) [10]. However, even though several studies have reported the safety of TNTs and their modified versions [7,8,9,10,61,85], a deeper toxicity assessment should be initiated at diverse levels (cytotoxicity, genotoxicity, carcinogenicity, etc.) to assure a smooth transformation of titanate nanotubes into therapeutic practice, and to provide a guarantee for patients/markets that these nanocarriers are as safe as possible. 

In a nutshell, we believe that the optimization of hydrothermal synthesis of TNTs and their surface modification present a wide window for future research to improve their properties and avoid their obstacles (enhancing pharmacokinetics and assuring safety). This would promote their pioneering application as a drug carrier and write a new chapter in the field of nano-based drug delivery systems.

## 7. Conclusions

TNTs are emerging as an attractive candidate to apply in health-related applications after proving their impressive performance in numerous disciplines. Therefore, this review attempts to highlight the importance of establishing a reproducible method for the preparation of TNTs with predesigned specifications that could fit the strict pharmaceutical requirements. This cannot be achieved without a comprehensive understanding of the factors affecting their preparation and the impact of its variation on the resulting nanotubes.

During hydrothermal treatment, several factors could be responsible for the final result. Temperature is one of the most important factors among them, as it is involved in the preparation and post-preparation treatment to provide sufficient thermal energy for rolling up and controlling the crystalline structure, respectively. This would not be possible without the use of a highly concentrated alkaline medium in addition to a sufficient time for the reaction to be completed. It is also worth mentioning that the properties/quantities of the starting materials also have a huge impact on the reaction pathway and its final product. Washing with acid as a post treatment step also has significant importance, but its role is still generating arguments and is considered as a point of disagreement. Based on the available data, the determination of the impact of preparation conditions on the resulting nanotubes is still considered to be challenging. This is crucially important as these operational conditions closely affect the properties of the produced nanotubes and therefore their subsequent applications.

Once the exact mechanism of TNTs production and the effect of the preparation parameters on their formation are clearly understood, a reproducible fabrication procedure could be generated and scaled up to create controlled features of TNTs that could fit their intended future implementation in therapeutic systems as drug carriers.

## Figures and Tables

**Figure 1 pharmaceutics-16-00635-f001:**
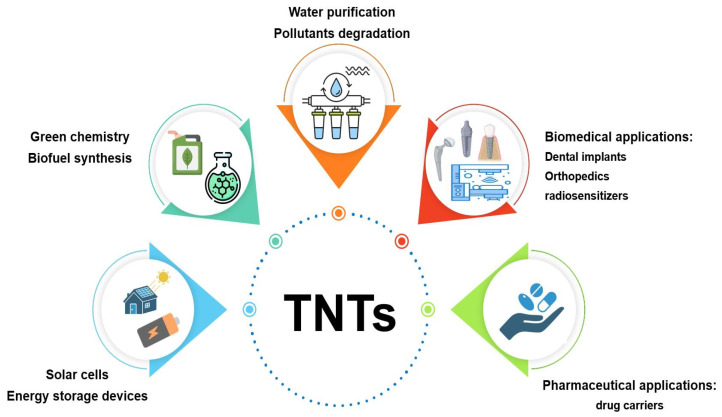
The applications of TNTs in different platforms.

**Figure 2 pharmaceutics-16-00635-f002:**
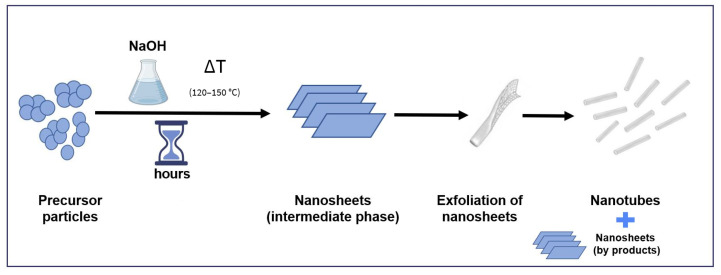
Schematic illustration of the mechanism of nanotubes formation during hydrothermal treatment method.

**Figure 3 pharmaceutics-16-00635-f003:**
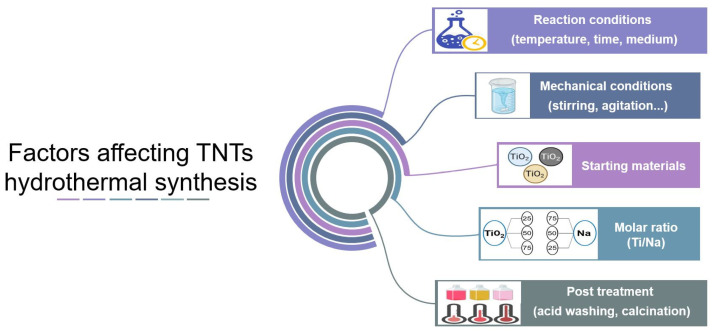
Factors affecting TNTs synthesis by hydrothermal treatment.

**Table 1 pharmaceutics-16-00635-t001:** Some of the reported biomedical/pharmaceutical application of TNTs prepared by several methods other than hydrothermal treatment.

Synthesis Method	Diameters (nm)	Loaded Agent	Application	Ref.
anodization	80	Bovine serum albumin/lysozyme	Controlled drug release in orthopedics and dental implants	[65]
anodization	80	Gentamicin	- Local delivery of antibiotics at the site of implantation - Minimize initial bacterial adhesion - Enhance osteoblast differentiation	[66]
anodization	60	Penicillin Streptomycin Dexamethasone	Prolonged drug release up to 3 days (nanotubes were coated with drugs)	[67]
anodization	-	Violet-blue fluorescent marker	Magnetically guided nanotubes for site-specific photoinduced drug release	[68]
anodization	100–300	Albumin Sirolimus Paclitaxel	Small and large molecules for controlled delivery	[69]
Sol gel/ anodization	130	Transcriptional factors	Protein delivery	[70]
anodization	110	Coumarin6 Indomethacin	Implantable extended drug delivery for poorly water-soluble drug	[71]
anodization	-	Indomethacin Itraconazole Gentamicin sulfate	Multi-drug delivery system with immediate, delayed and sustained therapeutic action possibility in bone implant	[72]
anodization	25–100	Silver NPs	Anti-bacterial activity against *E. coli*	[73]
anodization	180	Silver NPs	Anti-bacterial activity against *E. coli* and *S. aureus*	[74]
anodization	80	Silver NPs	Anti-bacterial activity	[75]
anodization	70–90	cefuroxime	reservoir on the orthopedic implant for local delivery	[76]
anodization	-	Gentamicin coated with biopolymers (PLGA and chitosan)	- Sustained release (3–4 weeks) - excellent osteoblastic adhesion - effective antibacterial properties	[77]
anodization	70	alendronate	titanium-based implant with local anti-osteoporosis property	[78]
Sol-gel	10–38	5-Fluorouracil	anticancer sustained drug delivery system	[79]

## Data Availability

The data used in the current review are available in public databases.

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
