# Peer review of "An Overview of Hydrothermally Synthesized Titanate Nanotubes: The Factors Affecting Preparation and Their Promising Pharmaceutical Applications"

_pharmaceutics, 2024, doi:10.3390/pharmaceutics16050635_

Round 1

Reviewer 1 Report

Comments and Suggestions for Authors

The review manuscript presented by Saker et al. summarizes the different synthesis and potential drug delivery applications of hydrothermally synthesized TiO2 nanotubes. The review document is interesting, and its information shows important literature collections that illustrate the applicability of these nanostructures. However, after a careful revision, important issues are recommended.

The review must contain more discussion about pharmaceutical applications of TNTs synthesized by hydrothermal method.

The review must include more illustrations. Moreover, the authors are invited to include relevant results from outstanding works with the appropriate citations.

The introduction must be improved by focusing on paragraph transitions.

Specify which parameter is discussed in lines 41-52. Nanotube diameter, nanotube thickness, nanotube length?

Replace the word study by review in the introduction (lines 53, 93).

Please include the design methodology used for selecting and evaluating the articles used in this review.

Please highlight the specific objective of the work and the key considerations to be highlighted in this review.

Change the position of the sentence in line 53. It could be in the last paragraph of the introduction.

Improve Figure 1, focusing on the different conditions, parameters, and results of the nanotubes.

Please include a section regarding the advantages of the hydrothermal method for the synthesis of nanotubes. Also, consider improving the conclusion by considering new insights into the application of hydrothermal synthesis methods.

Comments on the Quality of English Language

English needs to be improved. 

Reviewer 2 Report

Comments and Suggestions for Authors

Here, TNTs preparation and pharmaceutic applications are briefly summarized. Overall, this review could provide opinions on establish producible methods for TNTs production and help optimize their characteristics toward pharmaceutical industry and drug delivery. A few concerns are suggested to be solved during revision.

1. In Table 1, more parameters, including reaction temperature, raw materials, reaction time, reaction pressure, etc., should be appropriated added.

2. Apart from the descriptions in the main text, the reactions conditions for TNTs preparation should also be summarized by creating the corresponding tables.

3. How do these reaction conditions affect the TNTs morphological features and physiochemical properties? These reaction mechanisms should be discussed at the end of each sub-sections.

4. A major concern of TNTs for applying in pharmaceutical field is their blood degradation and biological safety. These points of focus is worth discussion in section 5.

5. Future research direction of TNTs preparation and pharmaceutical application need to be overviewed in detail.

Reviewer 3 Report

Comments and Suggestions for Authors

The authors present a review on hydrothermally synthesized TNTs and discusses the factors that affect their synthesis and their pharmaceutical applications. The revision could help to gain deeper knowledge and answer to several problems concerning establishing a producible method of TNTs production. In addition, it could also help to optimize the characteristics of TNTs and broaden their application, especially in the pharmaceutical industry and drug delivery. However, major revisions are needed for the paper to be suitable for publication in Pharmaceutics.

1. Introdcution section: authors should define TNTs and discuss their structure and properties.

2. References are needed in the paragraph from line 35 to line 40.

3.  Applications in pharmaceutical field section: the references used in the TNT applications are somewhat outdated. The authors should use more current references. It makes no sense to use non-current references in a review that is intended to reflect the state of the art.

4. Conclusions section: the authors should add some conclusions on the applications of TNTs.

Minor points

1.       The figures in the paper are of poor quality. The authors should improve them.

2.       Figure 1 should be mentioned in the paragraph from line 53 to line 74.

Round 2

Reviewer 1 Report

Comments and Suggestions for Authors

The authors have improved their work after following the recommendations of the reviewers. 

Comments on the Quality of English Language

The draft still requires English improvement.

Reviewer 2 Report

Comments and Suggestions for Authors

Acceptance can be considered after this revision. 

Reviewer 3 Report

Comments and Suggestions for Authors

The authors have improved the paper taking into account the reviewer's suggestions. In my opinion, the revision in its current version is suitable for publication in Pharmaceutics.